# Why urban communities from low-income and middle-income countries participate in public and global health research: protocol for a scoping review

Marie-Catherine Gagnon-Dufresne [1,2] Ivan Sarmiento [3,4] Geneviève Fortin,[1,2] Neil Andersson,[3,5] Kate Zinszer[1,2]

¹Department of Social and Preventive Medicine, University of Montreal School of Public Health, Montréal, Quebec, Canada
²Centre de recherche en santé publique (CReSP), Montréal, Quebec, Canada
³Department of Family Medicine, McGill University, Montréal, Quebec, Canada
⁴Escuela de Medicina y Ciencias de la Salud, Universidad Del Rosario, Bogota, Colombia
⁵Centro de Investigación de Enfermedades Tropicales (CIET), Universidad Autonoma de Guerrero - Campus Acapulco, Acapulco, Guerrero, Mexico

**Correspondence to**
Dr Ivan Sarmiento;
ivan.sarmiento@mail.mcgill.ca

## ABSTRACT

**Introduction** As the number of people living in cities increases worldwide, particularly in low-income and middle-income countries (LMICs), urban health is a growing priority of public and global health. Rapid unplanned urbanisation in LMICs has exacerbated inequalities, putting the urban poor at increased risk of ill health due to difficult living conditions in cities. Collaboration with communities in research is a key strategy for addressing the challenges they face. The objective of this scoping review is, therefore, to identify factors that influence the participation of urban communities from LMICs in public and global health research.

**Methods and analysis** We will develop a search strategy with a health librarian to explore the following databases: MEDLINE, Embase, Web of Science, Cochrane, Global Health and CINAHL. We will use MeSH terms and keywords exploring the concepts of 'low-income and middle-income countries', 'community participation in research' and 'urban settings' to look at empirical research conducted in English or French. There will be no restriction in terms of dates of publication. Two independent reviewers will screen and select studies, first based on titles and abstracts, and then on full text. Two reviewers will extract data. We will summarise the results using tables and fuzzy cognitive mapping.

**Ethics and dissemination** This scoping review is part of a larger project to be approved by the University of Montréal's Research Ethics Committee for Science and Health in Montréal (Canada), and the Institutional Review Board of the James P Grant School of Public Health at BRAC University in Dhaka (Bangladesh). Results from the review will contribute to a participatory process seeking to combine scientific evidence with experiential knowledge of stakeholders in Dhaka to understand how to better collaborate with communities for research. The review could contribute to a shift toward research that is more inclusive and beneficial for communities.

## INTRODUCTION

As the number of people living in cities increases worldwide, particularly in low-income and middle-income countries (LMICs), the health of urban populations

### STRENGTHS AND LIMITATIONS OF THIS STUDY

⇒ What constitutes 'communities', 'participation in research' and 'urban settings' can be defined in various ways, so it will be crucial to highlight how these concepts are defined in the literature included in the scoping review.
⇒ Reporting on community participation is heterogeneous, and identifying the research approaches, health issues, contexts and community characteristics that favour participation will be challenging.
⇒ The scoping review will summarise results using fuzzy cognitive mapping, providing soft models of causality that can be contextualised in the experience of local stakeholders in Dhaka.
⇒ The methods presented in this scoping review protocol could be replicated to compare and combine scientific evidence and experiential knowledge anywhere.

is a growing priority of public and global health.[1] Urbanisation brings changes to the disease burdens, determinants of health, and patterns of health inequalities.[1 2] Despite the benefits of urban living and progress in population health, rapid and unplanned urbanisation in LMICs has worsened health inequalities.[3] Not everyone in cities experiences these improvements equally, as policies and other efforts often fail to reach the most marginalised communities, including those living in informal settlements.[3] The urban poor are, therefore, at increased risk of ill health due to the difficult living conditions in cities.[3 4]

Several researchers in public and global health have criticised the reproduction of colonial relations in efforts to improve the health of populations in LMICs, as these efforts are often led by foreign researchers with little input from local populations.[5–7] The resulting unequal power dynamics between researchers and communities are

among the reasons research makes little or mixed contributions to health.[8] In response, there are increasing calls to decolonise public and global health through community participation in research, to better meet their needs and ensure local relevance of the initiatives put in place to improve their health.[8–10]

Community participation in health research is recognised for building capacity and fostering conditions to enable better community control over determinants of their health.[11 12] Community participation can lead to equitable partnerships between communities and researchers, making research more empowering and effective.[12] However, most health research uses top-down community engagement approaches rather than bottom-up participatory methods.[13] While there is no standard definition of community participation, different uses of the term form a continuum from consulting or informing communities to sharing power with them.[13 14]

Some of the world's most populated cities are located in South Asian countries, including Bangladesh, India and Pakistan.[15] These countries are characterised by high levels of urban poverty, with more than 50% of their urban population estimated to be living in informal settlements.[16 17] Considering that urbanisation in these contexts is inextricably linked to complex patterns of discrimination and social exclusion for residents of informal settlements, it is crucial that public and global health professionals and researchers collaborate with these communities to understand their health priorities and find innovative solutions to improve their health.[18 19]

Collaborating with marginalised urban communities is a key strategy for addressing the many challenges they face.[3 20 21] Yet, these communities represent a particularly hard-to-reach group, as asymmetries in access to resources and opportunities affect their capacity to fully participate in and benefit from research implemented to improve their health.[20 22] There is, therefore, an urgent need to better understand the barriers and enablers to their participation.[23–25]

The objective of this scoping review is to identify factors that influence the participation of urban communities from LMICs in public and global health research. Part of a larger project, this scoping review will contribute to a dialogue between scientific and experiential knowledge on the factors that influence community participation in public and global health research. We will contextualise the results from the scoping review in the views of stakeholders in Dhaka (Bangladesh) in a participatory process to reflect their experiences. This contextualisation will identify barriers and enablers to participation that are specific to Dhaka, in preparation for a cluster randomised controlled trial testing the effect of a participatory community mobilisation intervention for reducing dengue infection.[26]

## METHODS AND ANALYSIS

This review is part of a larger project aimed at comparing and combining different knowledge sources to provide a knowledge base for decision-making, which will be used to inform an upcoming cluster randomised controlled trial to reduce dengue infection in Dhaka (trial registration number: ISRCTN66131315).[26] The larger project will consider four knowledge sources: (1) the scoping review described in this protocol; (2) the views of Canadian and Bangladeshi public and global health researchers; (3) the views of personnel from community-based organisations in Dhaka and (4) the views of community members from underserved neighbourhoods in Dhaka. We will adapt the "Weight of Evidence" approach[27 28] and use fuzzy cognitive mapping (FCM) to bring these different knowledge sources into conversation.

In this protocol, we focus on describing the procedures to conduct the scoping review and briefly discuss how the results will be used to inform the subsequent phases of the larger project. The proposed scoping review will be conducted in accordance with the Preferred Reporting Items for Systematic Reviews and Meta-Analyses—Extension for Scoping Reviews (PRISMA-ScR) guidelines and the Joanna Briggs Institute's methodology to ensure accuracy, completeness and transparency.[29–32]

### Review questions

This scoping review will seek to answer the following question, developed according to the Participants, Concept, Context (PCC) method recommended by the Joanna Briggs Institute[29]: what factors influence the participation of urban communities from LMICs in research, based on evidence from the public and global health literature?

The review will also seek to answer the following subquestions:
1. What are the main barriers and enablers of community participation in public and global health research?
2. What is the relative influence of these factors on community participation?
3. What research approaches are most and least favourable to community participation?
4. What public and global health issues are most and least favourable to community participation?
5. What contexts are most and least favourable to community participation?
6. What community characteristics are most and least favourable to community participation?

Table 1 summarises the eligibility criteria for the scoping review following the PCC method. These criteria will be explained in more detail in the next section.

### Eligibility criteria
#### Types of sources

This scoping review will consider empirical studies with quantitative, qualitative and mixed methods designs. We will not include literature reviews and meta-analyses, but we will consider including the empirical studies reported in reviews and meta-analyses if relevant. We

**Table 1** Eligibility criteria for the scoping review, based on the PCC method

| PCC element | Correspondence in the scoping review |
| --- | --- |
| Participants | Communities in urban settings:<br>▶ Communities are defined as groups of individuals linked by shared social ties or interests who engage in joint actions[33–35]<br>▶ Urban settings, as defined by the authors of the articles included in the scoping review |
| Concept | Community participation in research:<br>▶ Research involving non-academic stakeholders in decision-making over some aspects of the research[14] |
| Context | LMICs:<br>▶ Countries included in the World Bank's classification of LMICs based on gross national incomes per capita[41] |

LMICs, low-income and middle-income countries; PCC, Participants, Concept, Context.

will not consider grey or scientific literature reporting on programmes, policy or other initiative implemented outside of research purposes, since our focus is community participation in research.

### Participants

Participants for this review will be communities in urban settings. Because this review will look at research participation at the community level rather than at the individual level, we will exclude studies discussing the participation of individuals in research (i.e., patient engagement and individual motivation to participate). The term 'community' generally refers to population groups and the locus (i.e., place, venue or other units) of their actions.[33] For this scoping review, we define communities as groups of people with diverse characteristics that are linked by social ties or identities; share common interests or concerns; and engage in joint action in settings, venues or areas that may be physically, geographically, culturally or politically defined.[33–35] The definition of what constitutes a community will, therefore, remain broad for this scoping review to ensure that we consider all relevant studies discussing the participation of communities in public and global health research.

We will, however, focus on communities located in urban settings, excluding rural communities. There is no standard international definition of what constitutes an urban setting. Each country has its own definition, following nationally defined criteria on population size, population density, type of economic activity, physical characteristics, level of infrastructure or other characteristics.[36] Considering the lack of a common definition, the scoping review will consider all studies in LMICs conducted in urban settings or cities as identified by the authors, including neighbourhoods and informal settlements (slums) in cities.

Our focus on urban communities in LMICs is justified by the fact that the larger project is part of a cluster randomised controlled trial on dengue, which will be conducted in Dhaka (Bangladesh), and that the factors influencing community participation in research will likely vary between rural and urban communities, and between high-income countries and LMICs.

### Concept

Community-engaged research is a broad topic, defined in various ways and used for numerous reasons. It is often an umbrella term for research involving the participation of non-academic stakeholders, with diverse models and conceptual frameworks.[14] There is no standard definition of community participation in research in public and global health.[13 14] The distinction between 'engagement,' 'mobilisation' and 'participation' in research is unclear, as these terms are often used interchangeably and with changing definitions.

Various authors discuss the levels of community participation in research as being positioned along a continuum, ranging from information provision and exchange, to consultation, to co-production and to shared leadership and community control.[14 37–40] For this scoping review, we will not restrict the search to a specific level of participation, but we will examine and compare how different approaches (i.e., community mobilisation interventions, partnered research, community-based participatory research designs, etc) are found to enable or hinder participation. However, research in which there is little community involvement (i.e., health education and consultation efforts in which communities have no decision-making power over some aspects of the research) will be excluded.

### Context

This review will focus on LMICs. The definition of LMICs used for the review is based on the World Bank's classification from the 2023 fiscal year, established following a country's gross national income per capita.[41] The Cochrane Effective Practice and Organisation of Care group has developed a filter for literature reviews based on the World Bank's classification to identify studies relevant to LMICs.[42]

### Exclusion criteria

To ensure the selection of relevant studies for the review, we will use the following exclusion criteria:
▶ Grey literature (institutional reports from non-governmental organisations, policy documents or other documents not reporting on research projects).
▶ Not empirical research.
▶ Discussing community engagement, participation, partnership or mobilisation in contexts other than research (i.e., programmes, policy and urban planning).

**Table 2** Inclusion and exclusion criteria for the scoping review

| Inclusion criteria | Exclusion criteria |
|---|---|
| ► Empirical qualitative, quantitative and mixed methods research<br>► Discussing community engagement, participation, partnership or mobilisation in research<br>► Focused on community-level engagement<br>► Discussing factors that influence community participation in research<br>► Conducted in urban settings<br>► Conducted in LMICs | ► Not empirical research<br>► Grey literature, including reports from non-governmental organisations or policy documents<br>► Discussing community engagement, participation, partnership or mobilisation in contexts other than research<br>► Focused on individual-level engagement or on the individual experiences of participants<br>► Not discussing factors that influence community participation in research<br>► Conducted in contexts other than urban settings<br>► Conducted in countries other than LMICs<br>► Full text of the reference not available |

LMICs, low-income and middle-income countries.

► Reports on individual engagement in research (i.e., patient engagement and individual motivation) or on the individual experiences of participants.
► Not discussing factors that influence community participation in research.
► Conducted in contexts other than urban settings.
► Conducted in countries other than LMICs.
► Full text of the reference not available.

Table 2 summarises the inclusion and exclusion criteria used to select articles for the scoping review.

## Search strategy

The search strategy will be developed with the help of a health librarian. It will explore the following databases: MEDLINE, Embase, Web of Science, Cochrane, Global Health and CINAHL. We will use MeSH terms and keywords to identify studies reported in English or French. We will not have restrictions in terms of dates of publication. We will not contact the authors of the articles selected to request additional information. Box 1 presents the initial search strategy for MEDLINE, which will be adapted for each database.

The search strategy will be developed with the input from a librarian and the research team to identify new keywords.[43] After our initial screening in MEDLINE, we will search the included articles for new keywords. A new search will then be conducted combining the newly found MeSH terms and keywords to the existing search. A librarian will assess whether these new terms should be included in the final search strategy. When all articles are screened, we will search the reference lists of selected

**Box 1   Example of a potential search strategy for MEDLINE**

**Concept 1: LMICs**
1. (afghanistan or albania or algeria or american samoa or angola or 'antigua and barbuda' or antigua or barbuda or argentina or armenia or armenian or aruba or azerbaijan or bahrain or bangladesh or barbados or 'republic of Belarus' or belarus or byelarus or belorussia or byelorussian or belize or british honduras or benin or dahomey or bhutan or bolivia or 'bosnia and herzegovina' or bosnia or herzegovina or botswana or bechuanaland or brazil or brasil or bulgaria or burkina faso or burkina fasso or upper volta or burundi or urundi or cabo verde or cape verde or cambodia or kampuchea or khmer republic or cameroon or cameron or cameroun or central african republic or ubangi shari or chad or chile or china or colombia or comoros or comoro islands or iles comores or mayotte or 'democratic republic of the congo' or democratic republic congo or congo or zaire or costa rica or 'cote d'ivoire' or 'cote d' ivoire' or cote divoire or cote d ivoire or ivory coast or croatia or cuba or cyprus or czech republic or czechoslovakia or djibouti or french somaliland or dominica or dominican republic or ecuador or egypt or united arab republic or el salvador or equatorial guinea or spanish guinea or eritrea or estonia or eswatini or swaziland or ethiopia or fiji or gabon or gabonese republic or gambia or 'georgia (republic)' or georgian or ghana or gold coast or gibraltar or greece or grenada or guam or guatemala or guinea or guinea bissau or guyana or british guiana or haiti or hispaniola or honduras or hungary or india or indonesia or timor or iran or iraq or 'isle of man' or jamaica or jordan or kazakhstan or kazakh or kenya or 'democratic people's republic of korea' or 'republic of korea' or north korea or south korea or korea or kosovo or kyrgyzstan or kirghizia or kirgizstan or kyrgyz republic or kirghiz or laos or lao pdr or 'lao people's democratic republic' or latvia or lebanon or lebanese republic or lesotho or basutoland or liberia or libya or libyan arab jamahiriya or lithuania or macau or macao or 'republic of north macedonia' or macedonia or madagascar or malagasy republic or malawi or nyasaland or malaysia or malay federation or malaya federation or maldives or indian ocean islands or indian ocean or mali or malta or micronesia or 'federated states of Micronesia' or kiribati or marshall islands or nauru or northern mariana islands or palau or tuvalu or mauritania or mauritius or mexico or moldova or moldovian or mongolia or montenegro or morocco or ifni or mozambique or portuguese east africa or myanmar or burma or namibia or nepal or netherlands antilles or nicaragua or niger or nigeria or oman or muscat or pakistan or panama or papua new guinea or new guinea or paraguay or peru or philippines or philipines or phillipines or phillippines or poland or 'polish people's republic' or portugal or portuguese republic or puerto rico or romania or russia or russian federation or ussr or soviet union or 'union of soviet socialist republics' or rwanda or ruanda or samoa or pacific islands or polynesia or samoan islands or navigator island or navigator islands or 'sao tome and principe' or saudi arabia or senegal or serbia or seychelles or sierra leone or slovakia or slovak republic or slovenia or melanesia or solomon island or solomon islands or norfolk island or norfolk islands or somalia or south africa or south sudan or sri lanka or ceylon or 'saint kitts and nevis' or 'st. kitts and nevis' or saint lucia or 'st. lucia' or 'saint vincent and the grenadines' or saint vincent or 'st. vincent' or grenadines or sudan or suriname or surinam or dutch guiana or netherlands guiana or syria or syrian arab republic or tajikistan or tadjikistan or tadzhikistan or tadzhik or tanzania or tanganyika or thailand or siam or timor leste or east timor or togo or togolese republic or tonga or 'trinidad and tobago'

### Box 1  Continued

or trinidad or tobago or tunisia or turkey or turkmenistan or turkmen or uganda or ukraine or uruguay or uzbekistan or uzbek or vanuatu or new hebrides or venezuela or vietnam or viet nam or middle east or west bank or gaza or palestine or yemen or yugoslavia or zambia or zimbabwe or northern rhodesia or global south or 'africa south of the sahara' or sub-saharan africa or subsaharan africa or africa, central or central africa or africa, northern or north africa or northern africa or magreb or maghrib or sahara or africa, southern or southern africa or africa, eastern or east africa or eastern africa or africa, western or west africa or western africa or west indies or indian ocean islands or caribbean or central america or latin america or 'south and central america' or south america or asia, central or central asia or asia, northern or north asia or northern asia or asia, southeastern or southeastern asia or south eastern asia or southeast asia or south east asia or asia, western or western asia or europe, eastern or east europe or eastern europe or developing country or developing countries or developing nation? or developing population? or developing world or less developed countr* or less developed nation? or less developed population? or less developed world or lesser developed countr* or lesser developed nation? or lesser developed population? or lesser developed world or under developed countr* or under developed nation? or under developed population? or under developed world or underdeveloped countr* or underdeveloped nation? or underdeveloped population? or underdeveloped world or middle income countr* or middle income nation? or middle income population? or low income countr* or low income nation? or low income population? or lower income countr* or lower income nation? or lower income population? or underserved countr* or underserved nation? or underserved population? or underserved world or under served countr* or under served nation? or under served population? or under served world or deprived countr* or deprived nation? or deprived population? or deprived world or poor countr* or poor nation? or poor population? or poor world or poorer countr* or poorer nation? or poorer population? or poorer world or developing econom* or less developed econom* or lesser developed econom* or under developed econom* or underdeveloped econom* or middle income econom* or low income econom* or lower income econom* or low gdp or low gnp or low gross domestic or low gross national or lower gdp or lower gnp or lower gross domestic or lower gross national or lmic or lmics or third world or lami countr* or transitional countr* or emerging economies or emerging nation?).ti,ab,sh,kf.

2. Developing countries/.
3. 1 or 2.

**Concept 2: community participation in research**

4. (((participat* or communit* or partner*) adj3 research) or (communit* adj3 (participat* or engage* or mobili?ation or intervention*)) or participatory or CBPR).ti,ab,sh,kf.
5. Community-based participatory research/.
6. Community participation/.
7. 4 or five or 6.

**Concept 3: urban settings**

8. (urban* or city or cities or metropol* or megacit* or megalop* or municipalit* or "informal settlement" or "informal settlements" or slum* or favela* or "shanty town" or "shanty towns" or ghetto* or bustee*).ti,ab,sh,kf.
9. Urban Health/.
10. Urban Population/.

Continued

### Box 1  Continued

11. Cities/.
12. Urbanisation/.
13. Poverty Areas/.
14. 8 or nine or 10 or 11 or 12 or 13.

**Final search strategy**

15. 15.3 and 7 and 14.

LMICs, low-income and middle-income countries.

studies to identify additional studies meeting our inclusion criteria.

## Study selection

Following the search, all identified citations will be collated and uploaded into Covidence,[44] and duplicates will be removed. Study selection will be conducted in two phases by two independent reviewers, who will reconcile differences by consensus. A third independent reviewer will help resolve any further disagreement.

The initial screening of the retrieved sources will use titles and abstracts. The second phase of selection will use full text. Reasons for excluding sources that do not meet the inclusion criteria at full text will be recorded and reported in the review. The results of the search and the study selection process will be reported in the flow diagram developed by PRISMA-ScR.[32 45]

Because the aim of scoping reviews is to map the available evidence on a specific topic, we will not perform an assessment of the methodological quality or risk of bias of the articles included in the review.[29] However, the data extraction form will report the research design as well as the data collection and analysis methods of selected articles. This will allow us to dress a portrait of the available evidence on the factors influencing the participation of urban communities in research.

## Data extraction

Two reviewers will develop and pilot a data extraction form, and extract the data in Covidence.[44] The form will include:

1. Details on the study (title, names of the authors, year of publication, study objectives, research design, and data collection and analysis methods).
2. The country and the urban settings in which studies were conducted.
3. Characteristics of participating communities.
4. If available, the definitions of 'community' and 'urban setting' used by the authors.
5. The participation approach used and the extent of community participation.
6. The findings regarding the factors (barriers, enablers and other factors) influencing the participation of urban communities in public and global health research.
7. If available, the relative influence (qualitative or quantitative) of the factors identified on community participation.

8. If available, other relations among the factors identified, and their relative influence on community participation.

9. Explanation of the relationships between factors (quotes from the articles).

We will not systematically extract data on the results of the studies since this is outside the scope of the review objectives and research question.

The data extraction form will be piloted before beginning the study selection process with a random sample of five studies among all the studies to be reviewed. The pilot test will help identify missing data and will contribute to ensuring that the reporting of participation approaches and factors influencing community participation is coherent across studies and between the two reviewers. The data extraction form will be modified and revised as necessary, in an iterative manner, during the data extraction process. Modifications will be detailed in the report of the review.

Any disagreements on data extraction that arise between the two reviewers at the pilot or data extraction stages will be resolved by consensus, or by discussion with a third independent reviewer if necessary.

### Data analysis and presentation

The presentation of results will follow the PRISMA-ScR guidelines.[32] We will present the results in tables and use FCM to illustrate how the different factors identified influence community participation in research, adapting the "Weight of Evidence" approach.[27 46] A narrative summary will also accompany the tabulated and mapped results, describing how the results relate to the review objectives and questions.

FCM uses graph theory and fuzzy logic to generate soft models of how change could happen based on assumed causal relationships.[47–49] These soft models are illustrated through graphs called fuzzy cognitive maps (figure 1), which are used to represent assumed causal relationships between concepts.[47 50] The maps use nodes (factors affecting the issue) and edges (arrows representing the relationships between factors), weighted by the relative strength of their influence on the outcome of

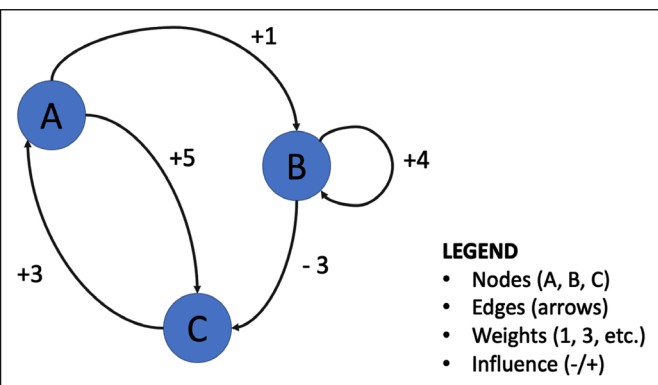

**Figure 1**  Example of a fuzzy cognitive map and associated concepts.

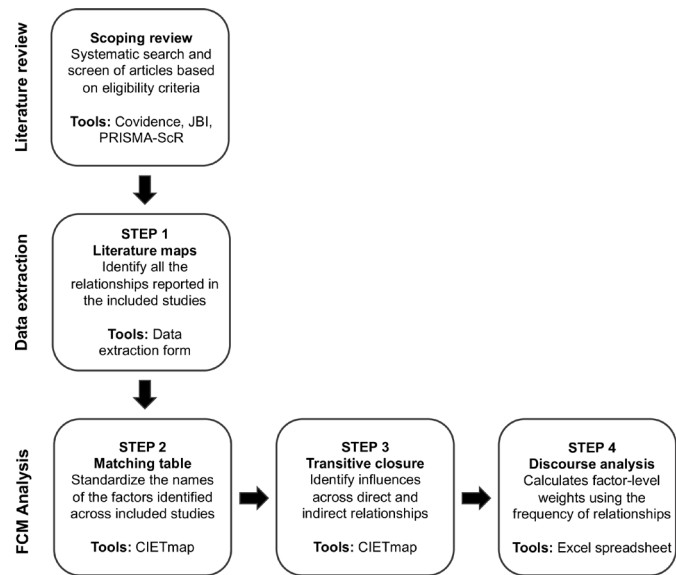

**Figure 2**  Steps of the FCM process for the scoping review. FCM, fuzzy cognitive mapping PRISMA-ScR, (Preferred Reporting Items for Systematic Reviews and Meta-Analyses—Extension for Scoping Reviews.

interest.[27 47 51] Depending on the knowledge source of the maps, edges can have different values (hence, the term fuzzy) to quantify their influence in a relative way.[49]

FCM will be the cornerstone for the presentation of the scoping review, through the creation of fuzzy cognitive maps to represent: (1) each article included in the review, and (2) a composite map for the whole review. FCM will allow to summarise in a composite map the relative influence that each factor might have on community participation, in relation to all the other factors identified in the review.[27 28 47] We will go through several steps (detailed below) to create the composite literature-based fuzzy cognitive map of the barriers and enablers to community participation (figure 2).

First, we will create one fuzzy cognitive map for each article selected in the scoping review (step 1 in figure 2). In each individual map, community participation will be the outcome of interest. We will include each barrier and enabler of community participation mentioned in the article (point f in the data extraction form) as a node in the map, which we will organise in a table. This table will have two initial columns indicating the origin factor (from) and the consequence factor (to). Additional columns will present the evidence supporting the relationship between both factors from the article (point i in the data extraction form). Each relationship identified will be a row in the table.[49]

Second, once all the individual tables are created, we will standardise the names of the factors across the individual articles so that they can be comparable (step 2 in figure 2).[46] For each individual map reporting the relationships identified in a study, we will calculate fuzzy transitive closure in the open access software CIETmap V.2.0 (step 3 in figure 2).[52] Fuzzy transitive closure is a mathematical model used to calculate the influence of each

relationship on community participation, considering all the possible relationships represented in the map.[53 54] After transitive closure, each relationship will have a value between 0 (having no influence) and 1 (having the strongest influence) to represent the relative strength of their influence on community participation, with positive and negative signs indicating whether the relationship is stimulative or inhibitive.[27 53]

Third, we will create a composite map for the whole review (step 4 in figure 2). To create this composite map, we will attribute weights to each relationship using Harris' discourse analysis. Harris' discourse analysis is an analytical approach developed in the 1950s based on the frequency of occurrence of discourse elements sharing similar meanings in a body of text (e.g., a literature review).[46 55] We will consider the frequency of occurrence of each relationship across all the individual maps developed for the studies included in the scoping review. This means that a factor that is repeated in multiple maps would have a stronger causal influence on community participation than a factor only mentioned in one or two maps.[46] We will establish the relative frequency of factors by dividing each occurrence by the highest frequency across all the maps. We will therefore obtain a value between 0 for the relationships that did not exist and 1 for the relationship most frequently mentioned.[46] These different steps will allow us to create a composite map representing all the factors and relationships identified in the scoping review, weighted according to their relative frequency.

### Patient and public involvement

We will include a consultation phase in the scoping review, as Arksey and O'Malley[56] recognised the benefit of discussing the results of a review with experts.[56] The "Weight of Evidence" approach, which we will adapt for this scoping review, advocates for experiential knowledge to be considered on an equal footing with the evidence synthesised from the literature.[27] Therefore, in the context of the larger project, which adopts a participatory methodology and involves a community advisory board, people concerned with the outcome of interest (i.e., participation of urban communities in health research) will be invited to contextualise the scoping review.

After conducting the scoping review, stakeholders in Dhaka will develop their own fuzzy cognitive maps on the factors that they believe can influence community participation in the Bangladeshi context. After creating their maps, they will interpret the literature-based map from the scoping review by comparing the results with their own maps. We will seek the perspectives of three stakeholder groups, namely, public and global health researchers, community-based organisations and community stakeholders.

Finally, we will use the composite map from the scoping review and the various maps from the three stakeholder groups to generate a final map incorporating these two knowledge sources. The literature-based map, the stakeholder maps and this final map will be reviewed through deliberative dialogue with stakeholders in Dhaka.[57] The maps and discussions with stakeholders will inform decision-making for the cluster randomised controlled trial on dengue testing a participatory community mobilisation intervention, where communities in Dhaka will be asked to develop their own solutions to reduce dengue infection. These steps will be conducted and reported separately.

## ETHICS AND DISSEMINATION

This scoping review does not require ethics approval. However, the consultation process is part of a larger project, which will need to be approved by the University of Montréal's Research Ethics Committee for Science and Health in Montréal (Canada), and the Institutional Review Board of the James P Grant School of Public Health at BRAC University in Dhaka (Bangladesh). We will apply for ethics approval for the larger project at both universities by August 2023. We will share the results from the scoping review with the scientific community through scientific articles and presentations at conferences, and with local stakeholders in Dhaka through a participatory process involving FCM and deliberative dialogue. Results from this process will directly inform the implementation of the cluster randomised controlled trial on dengue in Dhaka.[26]

## DISCUSSION

This protocol described a scoping review which will seek to identify and map the factors that can influence the participation of urban communities from LMICs in public and global health research. The review will contribute to the understanding of how to foster the participation of these communities in research, so that it can better respond to local needs. Given that marginalised urban communities represent a particularly hard-to-reach group in research and that urban health is a growing priority of public and global health, findings from this review will be useful for researchers and communities who wish to collaborate to improve population health.

The use of the "Weight of Evidence", an innovative approach to knowledge synthesis whereby scientific and experiential knowledge are brought into conversation, will allow for the contextualisation of the scoping review in the lived experience of stakeholders in Dhaka.[27 46 58] The procedures described in this scoping review protocol open the possibility for contextualising literature reviews in lived experience in any context.

One of the main challenges that we anticipate for the realisation of our scoping review is the time necessary to screen articles, as we expect that our search will yield a large number of studies. Discussions on the inclusion and exclusion criteria between the two reviewers and the research team prior to starting the screening process will contribute to ensuring our efficiency. We also recognise potential limitations of our scoping review. First, it is

possible that we miss studies that could have been relevant to our scoping review objectives if they were published outside the scientific literature (e.g., grey literature, reports from international or community organisations). Because we focus on articles written in English or French, we could also miss studies relevant to our objectives published in other languages. Our rigorous screening approach conducted by two independent reviewers will facilitate greater inter-reviewer reliability and maximise our chance of identifying all relevant studies. Second, the representation of the barriers and enablers of community participation as causal relationships through FCM is not meant to illustrate probability, but rather to represent soft models of causality that need empirical testing. In addition, our identification and classification of barriers and enablers of community participation rest on our subjective interpretation of the evidence. However, the use of FCM and Harris' discourse analysis to synthesise the results from the scoping review offers an operator-independent way to analyse and communicate the relative influence of the factors identified on community participation.[28] The literature-based map will in turn inform a mapping process involving stakeholders from Dhaka (Bangladesh), as part of the larger project. Third, we recognise that most research conducted in urban settings in LMICs focus on urban poor populations. It is, therefore, possible that most of the studies included in our review discuss underserved or marginalised populations, which is not necessarily representative of all communities living in cities in LMICs.

Better understanding the factors that influence the participation of communities in research could support a shift from researcher-driven health research toward research that is more inclusive of community voices and needs. Fostering authentic community participation in research can contribute to the movement for decolonising public and global health. This can also bring benefits to marginalised communities through interventions that are more relevant to their contexts and needs.

**Acknowledgements** The authors would like to thank the health librarians Sylvie Fontaine and Viviane Angers at the Université de Montréal for their support in developing this protocol. We would also like to thank members of the CIET/PRAM research lab at McGill University for sharing their expertise with us.

**Contributors** M-CG-D developed the scoping review protocol, the search strategy and wrote the first version of the manuscript. KZ and NA contributed to the development of the larger project to be conducted in Dhaka, as part of the COESA cluster randomised controlled trial. GF contributed to drafting the scoping review protocol. IS and NA provided expertise on fuzzy cognitive mapping, Harris' discourse analysis and the "Weight of Evidence" approach. All authors read, provided feedback and approved the final manuscript.

**Funding** This work was supported by the Fonds de recherche du Québec – Santé (FRQS) through a doctoral research scholarship awarded to the first author. This scoping review is part of a larger project supported by the Canadian Institutes of Health Research through the Project (grant program number: 201803PJT-400444-RC2-CFCA-120159). These institutions did not play a role in the development of this protocol.

**Competing interests** None declared.

**Patient and public involvement** Patients and/or the public were not involved in the design, or conduct, or reporting, or dissemination plans of this research.

**Patient consent for publication** Not applicable.

**Provenance and peer review** Not commissioned; externally peer reviewed.

**ORCID iDs**
Marie-Catherine Gagnon-Dufresne http://orcid.org/0000-0002-5100-6790
Ivan Sarmiento http://orcid.org/0000-0003-2871-1464

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
