## [Reviewer comments · BMJ Open]

ARTICLE DETAILS

TITLE (PROVISIONAL)	Why urban communities from low- and middle-income countries participate in public and global health research: Protocol for a scoping review
AUTHORS	Gagnon-Dufresne, Marie-Catherine; Sarmiento, Ivan; Fortin, Geneviève; Andersson, Neil; Zinszer, Kate

VERSION 1 – REVIEW

REVIEWER	Abboah-Offei, Mary University of York
REVIEW RETURNED	19-Jan-2023

GENERAL COMMENTS	This is a very interesting and informative protocol, and if duly carried out, will help us to understand why urban communities from low- and middle-income countries actually participate in public and global health research. Please review and revise these minor observations: 1. On page 10 line 30, you wrote "data extraction stages will be resolved through by consensus", could you clarify whether you meant to say "resolved through" or "resolved by"2. You indicated "However, the consultation process is part of a larger project which will need to be approved by the University of Montréal's Research Ethics Committee for Science and Health in Montréal (Canada), and the Institutional Review Board of the James P. Grant School of Public Health at BRAC University in Dhaka (Bangladesh)." Could you please clarify when this ethics approval will be sought?3. On page 14, You indicated Figures 1 and 2, except that there were no figures although these figures were later found on pages 19 and 20, please in future check your manuscript to ensure that figures are at the appropriate places as stated. Best wishes.
--

REVIEWER	Lahariya, Chandrakant G. R. Medical College, Community Medicine
REVIEW RETURNED	24-Jan-2023

GENERAL COMMENTS	1. In the abstract, the type of the study i.e., scoping review can be included as well. A mention of the keywords that will be used for the search of the literature can also be mentioned. Detailed description will follow in the main body of the manuscript.
--

	2. Consider literature reporting on programs, policy or other initiative implemented outside of research purposes as authors are mentioning that this review will look at research participation at the community level rather than at the individual level, therefore excluding studies discussing the participation of individuals in research- in participants section. - There are some good and indepth review on urban health, which provides useful insight on urban health and can be included in literature search. i.e. Lahariya C, Bhagwat S, Saksena P, Samuel R. Strengthening urban health for advancing Universal Health Coverage in India. J Health Management 2016; 18: 361-6. 3. 4. What definition of community has been used by authors for scoping review should be explained in the paper. 5. Was the grey research included? If not then mention it in protocol. How the research questions were made? PICO or PCC what method was used? Methodology for study selection has been well explained in detail, but can a target for number of studies to be selected be included? Discuss. 6. Provide a table for inclusion and exclusion criteria for the scoping review. 7. How the quality of the studies was assessed? What tool was used to check the quality of studies? 8. The data analysis part of FCM can be better illustrated using a flow diagram. 9. Reaching a conclusion without having results in the protocol is not necessary. Instead, what outcomes (primary and secondary) are to be found can be written. And even these can be brief only without going into in depth explanations. 10. The potential sources of bias in the review process are acknowledged and addressed. 11. Reaching a conclusion without having results in the protocol is not necessary. Instead, what outcomes (primary and secondary) are to be found can be written. And even these can be brief only without going into in depth explanations.
--	--

VERSION 1 – AUTHOR RESPONSE

Reviewer: 1 (Dr. Mary Abboah-Offei, University of York)

This is a very interesting and informative protocol, and if duly carried out, will help us to understand why urban communities from low- and middle-income countries actually participate in public and global health research.

Thank you for taking the time to review our work and for your comments.

Please review and revise these minor observations:

1. On page 10 line 30, you wrote "data extraction stages will be resolved through by consensus", could you clarify whether you meant to say "resolved through" or "resolved by"?
We have corrected this mistake. We meant to say, "resolved by consensus" (see page 11, line 324).

2. You indicated "However, the consultation process is part of a larger project which will need to be approved by the University of Montréal's Research Ethics Committee for Science and Health in Montréal (Canada), and the Institutional Review Board of the James P. Grant School of Public Health

at BRAC University in Dhaka (Bangladesh)." Could you please clarify when this ethics approval will be sought?

We have added a sentence clarifying that we will submit formal ethics applications to both universities by August 2023 (see page 13, lines 417-418).

3. On page 14, you indicated Figures 1 and 2, except that there were no figures although these figures were later found on pages 19 and 20, please in future check your manuscript to ensure that figures are at the appropriate places as stated.

We apologize for the confusion. The submission guidelines require that we provide the legends of all figures at the end of the main document, but that we upload the figures themselves as separate files.

Reviewer: 2 (Dr. Chandrakant Lahariya, G. R. Medical College)

1. In the abstract, the type of the study i.e., scoping review can be included as well. A mention of the keywords that will be used for the search of the literature can also be mentioned. Detailed description will follow in the main body of the manuscript.

The last sentence of the 'Introduction' section of the Abstract mentions the type of study: "the objective of this scoping review is [...]" (see page 3, lines 39-41). We added the three concepts used to search the MeSH and keywords for the scoping review, namely "low- and middle-income countries", "community participation in research", and "urban settings" (see page 3, lines 45-47). We have cut words from the Abstract to fit the 300-word limit.

2. Consider literature reporting on programs, policy or other initiative implemented outside of research purposes as authors are mentioning that this review will look at research participation at the community level rather than at the individual level, therefore excluding studies discussing the participation of individuals in research – in participants section.

Our response to this comment has two aspects.

First, our objective is to identify factors that influence the participation of urban communities in public and global health *research*. We therefore decided to focus on literature reporting on research projects. We consider literature on programs, policy, or other health-related initiatives outside of research beyond our scope. While a distinction between research and other initiatives could be a limitation in a general context, in the specific context of our scoping review (aiming to inform an upcoming major research project), this criterion will help us focus on what we are trying to understand for the project in question (barriers and enablers of community participation in research).

Secondly, we are interested in articles addressing factors influencing *community-level participation* rather than on those that only describe the profile of participants in research (e.g., sociodemographic characteristics of participants, individual motivations for taking part in research). Because we want to learn about the processes/mechanisms of community engagement in research, we will exclude articles that focus on the individual characteristics of research participants. We will note the reasons for excluding articles at full text. One such reason will be 'discussing participation only at the individual level'. We will provide the references for all excluded articles at full text and will clearly state the reason for exclusion. This will allow other researchers wanting to look at individual-level participation to consult our list of references.

3. There are some good and in-depth review on urban health, which provides useful insight on urban health and can be included in literature search (i.e. Lahariya C, Bhagwat S, Saksena P, Samuel R.

Strengthening urban health for advancing Universal Health Coverage in India. J Health Management 2016; 18: 361-6 PubMed .).

We thank you for the recommendation to enrich our literature review for this protocol. We have added information on urban health in Asian countries in the 'Introduction' section (see pages 4-5, lines 116-123). We did not reference the interesting article by Lahariya and colleagues in this manuscript because it does not discuss community participation for urban health. Its focus on universal health coverage, non-communicable diseases, and healthcare financing is outside the scope of this protocol. We will however consider including it in reporting our scoping review, as we will discuss the importance of urban health in South Asian countries such as India, Pakistan, and Bangladesh.

4. What definition of community has been used by authors for scoping review should be explained in the paper.

We recognize that 'community' is a capacious concept. The 'Participants' section of the protocol specifies our use of the term as "groups of people with diverse characteristics that are linked by social ties or identities; share common interests or concerns; and engage in joint action in settings, venues or areas that may be physically, geographically, culturally, or politically defined" (see page 6, lines 192-195). For purposes of our scoping review, our definition of 'community' is thus deliberately broad. We will include studies that discuss community participation, without focusing on limited definitions of 'community' or 'participation'. We will report the various definitions of 'community' used across all the studies to show the multiple definitions of the term. We included details on this in the 'Data extraction' section (see page 10, line 302).

5. Was the grey research included? If not, then mention it in protocol.

This is an important point that we should have mentioned in the manuscript. We will not include grey literature to this scoping review (see Point 2 above). Our focus is community participation in *research*. We will not include grey literature on programs, policies or other initiatives implemented by community organizations, international organizations and/or governmental institutions. This limits the scope of our review and, we recognize, the conclusions we can draw from it. We included this information in the 'Types of sources' section of the protocol (see page 6, line 184) and in the 'Exclusion criteria' section (see page 7, line 241).

6. How were the research questions made? PICO or PCC what method was used?

We used the "Participants, Concept and Context" (PCC) method recommended by the Joanna Briggs Institute's guidelines to formulate our main review question, which is: "What factors influence the participation of urban communities from low- and middle-income countries in research, based on evidence from the public and global health literature?" (see pages 5-6, lines 163-165). As presented in the 'Eligibility criteria' section (see pages 6-7, lines 179-236), we created our research question according to the participants, concept and context targeted by our review:

- Participants: Urban communities
- Concept: Community participation/engagement/mobilization in research
- Context: Urban settings in low- and middle-income countries

We added a sentence to discuss how the review question was generated (see page 5, lines 161-163).

7. Methodology for study selection has been well explained in detail, but can a target for number of studies to be selected be included? Discuss.

We expect a large number of articles for this review, but it is difficult to anticipate how many articles our search will yield and how many will be included in the final selection. We added details on this in the 'Discussion' section of our protocol (see page 13, line 435-437).

8. Provide a table for inclusion and exclusion criteria for the scoping review.

This is a great suggestion. Table 2 now presents the inclusion and exclusion criteria for the review (see pages 7-8, lines 252-255).

9. How the quality of the studies was assessed? What tool was used to check the quality of studies? Following the argument of Peters and colleagues (2020) in the Joanna Briggs Institute’s guidelines for scoping reviews, quality appraisal of selected studies is not required in a scoping review because of the expected heterogeneity of the sources assessed. Our goal is to map existing evidence on participation of urban communities in research, not to arrive at an estimate as one might do in a meta-analysis. We will, however, document details of research designs, data collection, and data analysis in the data extraction form. This will allow us to dress a portrait of the evidence included in the scoping review. We added this information in the ‘Study selection’ section (see page 10, lines 287-292).

10. The data analysis part of FCM can be better illustrated using a flow diagram. Figure 2 provides a flow diagram of the FCM process. This illustrates the steps detailed in the ‘Data analysis and presentation’ section of the protocol. The steps presented in Figure 2 are now clearly identified as such in the text (see pages 11-12, lines 352, 361-362, 363, 371).

11. Reaching a conclusion without having results in the protocol is not necessary. Instead, what outcomes (primary and secondary) are to be found can be written. And even these can be brief only without going into in depth explanations. We changed ‘Conclusion’ to ‘Discussion’ (see page 13, line 425). We provide information about the expected outcomes and impacts of the scoping review in this section.

12. The potential sources of bias in the review process are acknowledged and addressed. We have added information on the potential limitations of our review process in the ‘Discussion’ section (see pages 13-14 lines 435-453).

13. Reaching a conclusion without having results in the protocol is not necessary. Instead, what outcomes (primary and secondary) are to be found can be written. And even these can be brief only without going into in depth explanations. We address this in our response to point 11 above (same comment).

VERSION 2 – REVIEW

REVIEWER	Lahariya, Chandrakant G. R. Medical College, Community Medicine
REVIEW RETURNED	30-Mar-2023

GENERAL COMMENTS	1. Urban poor are the most of the study participants that constitute the study sample especially those conducted in LMICs, creating a divide in the inference reached as per the results. Will the study address this disparity in participation and ultimately the representation? This is not mentioned in the research question. 2. What definition of community has been used by authors for scoping review should be explained in the paper 3. Consider literature reporting on programs, policy or other initiative implemented outside of research purposes as authors are mentioning that this review will look at research participation at the community level rather than at the individual level, therefore excluding studies discussing the participation of individuals in research- in participants section.
--

	4. Was the grey research included? If not then mention it in protocol. 5. Eligibility criteria can be neatly summarized in the form of a table or a flow diagram for easy and quick go through. 6. How the research questions were made? PICO or PCC what method was used? 7. Provide a table for inclusion and exclusion criteria for the scoping review. 8. How the quality of the studies was assessed? What tool was used to check the quality of studies? 9. The potential sources of bias in the review process are acknowledged and addressed. 10. Can include the definition of urban area and urban slums if available in the studies and mentioned by the authors. 11. The steps of Fuzzy Cognitive Mapping (FCM) can be represented in a simplified and concise manner in the form of a flow diagram. 12. The study protocol cannot have discussion section unless preceded by the results. Hence, it might need modification. Instead, a section of possible outcomes expected out of the scoping review can be written.
--	---

VERSION 2 – AUTHOR RESPONSE

Reviewer: 2 (Dr. Chandrakant Lahariya, G. R. Medical College)

1. Urban poor are the most of the study participants that constitute the study sample especially those conducted in LMICs, creating a divide in the inference reached as per the results. Will the study address this disparity in participation and ultimately the representation? This is not mentioned in the research question.

The revised submission clarifies that improving our understanding of the communities participating in research is a major aim of the approach we propose. We definitely agree that some groups might be more represented and/or engaged in research than others, and it is an explicit task of the project to explore these issues. Specifically, sub-questions #5 and #6 (see page 5, lines 172-174) will shed light on this. The revised submission provides additional details about this in the section on data extraction (point c) (see page 10, line 319) and in the 'Discussion' section (see page 14, lines 479-482).

2. What definition of community has been used by authors for scoping review should be explained in the paper.

The 'Participants' section (see page 6, lines 197-203) presents the definition of 'community' used for the scoping review. It describes communities as "groups of people with diverse characteristics that are linked by social ties or identities; share common interests or concerns; and engage in joint action in settings, venues or areas that may be physically, geographically, culturally, or politically defined."

We deliberately use a broad definition of “community” in the scoping review to ensure we do not exclude studies relevant to our topic. The revised submission clarifies this in the ‘Participants’ section (see page 6, lines 200-203). We recognise and report on the different uses of the term “community” found in the literature in the scoping review (see ‘Data extraction’, point d on page 10, line 320).

3. Consider literature reporting on programs, policy or other initiative implemented outside of research purposes as authors are mentioning that this review will look at research participation at the community level rather than at the individual level, therefore excluding studies discussing the participation of individuals in research – in participants section.

The limited objective of our scoping review is to identify factors that influence the participation of urban communities in public and global health *research*. We recognise there is merit in understanding participation in programs, policy, and other initiatives, but these activities almost all carry short term direct benefits for participants. They are likely to involve quite different dynamics to participation in research, where benefits that accrue do so usually in the longer term. The revised submission adds details to the exclusion criteria and to Table 2 to make this clearer (page 7, lines 258-259, 263-264, 270-274).

4. Was the grey research included? If not then mention it in protocol.

With the limitation of our objectives to global health research, we opted not consider grey literature in this scoping review (‘Eligibility criteria’ section). The revised submission clarifies this in ‘Types of sources’ (see page 5, lines 187-189). We also note in the ‘Exclusion criteria’ section (see page 7, lines 258-259) that ‘grey literature’ is an exclusion criterion for our review. We further address this as a potential limitation of our scoping review in the ‘Discussion’ section (see page 13, lines 463-465).

5. Eligibility criteria can be neatly summarized in the form of a table or a flow diagram for easy and quick go through.

We thank you for this suggestion. The revised submission adds Table 1 to summarise the eligibility criteria (see page 5, lines 176-181). This complements Table 2 created in response to the previous round of reviewer comments. We also rearranged the information found in the ‘Eligibility criteria’ section to better fit the PCC method (see pages 5-6, lines 192, 205-212).

6. How the research questions were made? PICO or PCC what method was used?

We used the “Participants, Concept and Context” (PCC) method recommended by the Joanna Briggs Institute’s guidelines (see page 4, lines 158-162). Table 1 now explains this in more detail (see page 5, lines 176-181).

7. Provide a table for inclusion and exclusion criteria for the scoping review.

Table 2 presents the inclusion and exclusion criteria used for our scoping review (see page 7, lines 270-274). We adjusted the content of Table 2 to respond to additional comments on the eligibility criteria. As mentioned above, Table 1 now complements this information.

8. How the quality of the studies was assessed? What tool was used to check the quality of studies?

The revised submission clarifies that we used the Joanna Briggs Institute's guidelines for scoping reviews written by Peters and colleagues (2020) to develop our protocol. These guidelines propose the quality appraisal of selected studies is not required in a scoping review because of the expected heterogeneity of the sources assessed. Following Peters et al. (2020), we will not assess the quality of the selected studies in this scoping review. We provide details about this decision in the protocol (see page 10, lines 305-310).

9. The potential sources of bias in the review process are acknowledged and addressed.

We discuss the limitations of our review process in the 'Discussion' section. The revised submission adds information about how our interpretation and classification of barriers and enablers rest on our interpretation of the evidence (see pages 13-14, lines 472-477).

10. Can include the definition of urban area and urban slums if available in the studies and mentioned by the authors.

The revised submission clarifies that we will discuss the definitions of urban areas (including slums and informal settlements, if applicable) when reporting the findings from the scoping review. We reorganised the information on the definition of 'urban areas' we will use for the review (see page 6, lines 205-212) and added point d in the 'Data extraction' section to address this comment (see page 10, line 320).

11. The steps of Fuzzy Cognitive Mapping (FCM) can be represented in a simplified and concise manner in the form of a flow diagram.

The revised submission adjusts Figure 2 to represent the different steps of the FCM process in a simplified manner (see modified Figure 2, attached to the submission).

12. The study protocol cannot have discussion section unless preceded by the results. Hence, it might need modification. Instead, a section of possible outcomes expected out of the scoping review can be written.

We appreciate the logic expressed here. The editor advised us, however, to use 'Discussion' as title for this section in the previous round of comments.

VERSION 3 – REVIEW

REVIEWER	Lahariya, Chandrakant G. R. Medical College, Community Medicine
REVIEW RETURNED	25-Apr-2023
GENERAL COMMENTS	This can be accepted